# The Correlation between Interleukin 33 and Psoriasis: A Systematic Review and Meta-Analysis

**Keshav Kc [1], Hua Hu [1], Tilak Mahatara [2], Sunil Koirala [2], Samjhana Shrestha [3], Shiv K. Sharma [4,*], Xiangfeng Song [5] and Zhongwei Tian [1,*]**

1   Department of Dermatology, The First Affiliated Hospital of Xinxiang Medical University, Xinxiang 453003, China
2   School of Health and Allied Sciences, Faculty of Health Sciences, Pokhara University, Pokhara 33700, Nepal
3   Health Research Institute, Faculty of Health, University of Canberra, Bruce, ACT 2617, Australia
4   Department of Chemistry, Thomas More University, 333 Thomas More Pkwy, Crestview Hills, KY 41017, USA
5   Department of Immunology, Xinxiang Medical University, Xinxiang 453000, China
*   Correspondence: sharmas1@thomasmore.edu (S.K.S.); zhonwt@xxmu.edu.cn (Z.T.)

**Abstract:** Psoriasis is a common genetic autoimmune disorder with a global prevalence of 2–3%. The clear pathogenesis of psoriasis is not fully understood, but hyperproliferation and inflammation of the epidermis with marked infiltration of immune cells have been indicated in psoriasis with such cells producing different types of cytokines- interleukin. As such a new member of the IL-1 cytokine family, in some research, IL-33 has been linked with psoriasis showing high serum concentration of IL-33 in human psoriatic plaques compared to normal healthy skin. Despite this, the association between IL-33 and psoriasis is not clear. Herein, in this review, we aim to investigate the correlation between serum IL-33 levels and psoriasis. We conducted meta-analysis using fixed or random-effects models to calculate pooled standard mean differences. We found that the mean IL-33 serum levels were reported between 0.35 pg/mL to 586 pg/mL in the psoriatic group and 0 pg/mL to 87.7 pg/mL in the healthy control group. Out of five, four individual studies included in the analysis reported statistically significant differences in IL-33 levels, the pooled estimate (SMD = 0.340 95% CI: −0.308 to 0.988), however, did not indicate a significant relation between IL-33 and psoriasis. This analysis revealed no significant difference between serum IL-33 levels in the psoriatic population in comparison to healthy controls. This may be because we did not include any animal studies, lab-based studies, any other markers mixed together, or any other cases of diseases mixed together. However, further research is warranted to confirm the reported association as this analysis is limited by the low-quality and observational nature of the included studies.

**Keywords:** serum IL-33; psoriasis; systematic review; meta-analysis

## 1. Introduction

Psoriasis, a condition in which skin cells build up and form itchy & dry patches, has been recognized as one of the key global health problems by the World Health Organization (WHO). Given the disfiguring and disabling disease nature and associated stigmatization, psoriasis is known to cause immense negative impacts on the quality of life of the patients [1]. Although psoriasis is reported to affect around 2–3% of the world's population [2], its ubiquity varies across the countries with prevalence figures ranging from 0.51% to 11.43% [3,4]. Psoriasis, as such, is a fairly common, immune-mediated chronic inflammatory skin disease [5] with potentially systemic implications that extends beyond the skin, and nails, thus increasing the risk of non-communicable diseases and co-morbidities [6]. Psoriasis can affect all age groups, but it mainly appears in early adulthood as a genetic disorder that is environmentally triggered (stress, injury, trauma, infections, and specific medications) [2]. Manifesting in many different forms (plaque, guttate, flexural, pustular, erythrodermic), psoriasis commonly features hyperproliferation and inflammation of the

epidermis allowing marked infiltration of immune cells such as dendritic cells (DCs), T cells, neutrophils and macrophages [7]. This abnormal infiltration of immune cells is mediated by a network of different cytokines (CKs) and their interactions, which play key roles in pro-inflammatory and anti-inflammatory signaling pathways driving the pathogenesis of the disease [8,9]. As such, the role of Interleukins (ILs) has been increasingly studied as one of the key CKs influencing the pathogenesis of psoriasis [10]. Studies have shown over expression of pro-inflammatory ILs (IL-1, IL-6, IL-8, IL-12, IL-15, IL-17, IL-18, IL-19, IL-20, IL-22, IL-23) in the lesional skin and serum of the psoriatic patients [10]. In addition to these ILs that have been shown to be associated with psoriasis, many new ILs have been identified in the past few years following the development of new technologies [11]. One recently discovered interleukin is IL-33 which is a new addition to the IL-1 cytokine family [12]. Produced by epithelial, endothelial, and fibroblast-like cells in both normal and inflamed states, IL-33 binds with ST2 receptors and imparts various effects on mast cells, Treg cells, Th-2 cells and NKT cells to regulate immune cells and tissue responses [12].

In recent years, research have revealed that normal healthy skin has a low amount of lesional IL-33 concentration compared to human psoriatic plaque [7,13–15]. Further, the use of various interventions or therapeutic measures such as methotrexate (MTX), narrowband ultraviolet B radiation (NB-UVB) and TNF-α inhibitors reported a change in the concentration of IL-33 in both lesion and serum [16]. For instance, the expression of IL-33 mRNA and protein after the treatment was decreased in psoriatic patients receiving MTX while it was increased in patients treated with NB-UVB [16]. Likewise, Mitsui et al. demonstrated a correlation between serum TNF-α and serum IL-33 levels and reported a decline in the elevated serum IL-33 levels in psoriatic patients after their treatment with anti-TNF-α therapy [7]. Furthermore, in Kobners reaction, healthy skin is manipulated in psoriatic patients and IL-33 was found to be correlated in such cases [17]. The existing evidence on the role of IL-33 and psoriasis is inconsistent as some studies have reported increased serum IL-33 levels in psoriasis patients compared to those with normal healthy skin [7,13–15], and some reported equal levels [9,18] while some reported lower serum IL-33 expression in patients with psoriasis [8]. For instance, Bodoor et al. found lower serum IL-33 levels in psoriasis patients compared to controls [8]. Although studies have demonstrated the role of IL-33 in the pathogenesis of psoriasis, suggesting its potential implications in therapeutics, only one literature review has been conducted so far to investigate the association between IL-33 and psoriasis [19]. This 2019 review that included 19 studies, however, did not provide the estimates of such association as the analysis was limited to narrative literature synthesis only [19]. IL-33 is involved in other systemic disease conditions such as allergic diseases, rheumatism, IBD, SLE, Alzheimer's disease and so on (Table 1). Since IL-33 is also detected in RA, IBD and other disease conditions, it would be easier to accept the possibility of its involvement in psoriasis. Patients with diabetes, habits of smoking, and alcohol consumption, etc. are also associated with serum IL-33 level. Smoking, for example, increases IL-33 in airway cells. Psoriasis is a disease in which disease activity is influenced by lifestyle. It may be necessary to further take into account the background factors such as smoking, obesity, and hyperlipidemia in the analysis and decreases IL-33 in the serum [20].

Provided that the clinical evidence regarding the role of IL-33 is inconclusive, and no systematic review has been performed yet to assess the correlation between IL-33 and psoriasis, we conducted a systematic review and meta-analysis to determine if IL-33 serum levels among psoriatic patients are similar to healthy controls and whether IL-33 serum levels have a significant relationship with disease severity.

**Table 1.** Studies in serum IL-33 in inflammatory and noninflammatory disorders. Reprinted with permission from J. Circulating Biomarkers, 2021.

| IL-33 Elevation | IL-33 Suppression | Constant IL-33 |
|---|---|---|
| Biliary atresia | HT of AIS | Heart failure |
| CKD | Poststroke depression | CKD |
| Rheumatoid arthritis | Long-term outcome after stroke | Stable angina pectoris, NSTEMI, STEMI |
| Rheumatoid arthritis | Intracerebral hemorrhage | NSTEMI |
| Psoriasis vulgaris | HF-REF | |
| Prostate cancer | Behçet's disease | |
| Gastric cancer | Critically ill subjects | |
| Endometrial cancer | Osteoporosis | |
| Non–small cell lung cancer | Amyotrophic lateral sclerosis | |
| Breast cancer | Increased body weight | |
| Sepsis in infants | | |

## 2. Materials and Methods

### 2.1. The Literature Searches and Study Selection

We conducted a systematic literature search of electronic databases including the Cochrane Central Registry of Controlled Trials (CENTRAL), and Clinical Trials Registries using the search terms: Interleukin 33 OR IL-33 or IL33 and Psoriasis, Palmo-plantaris Pustulosis or Pustular Psoriasis of Palms, Soles or Pustulosis Palmaris et Plantaris or Pustulosis of Palms, and Soles or Plaque psoriasis or Guttate psoriasis or Inverse psoriasis or Pustular psoriasis or Erythrodermic psoriasis. We limited our literary search to humans only. Furthermore, we also contacted professionals in the relevant field. The inclusion criteria were: (a) Population: adults (aged $\geq$ 18 years) and children (<18 years); (b) Disease: Psoriasis; (c) Comparator/Control: healthy adults and children; (d) Outcome measure: serum IL-33. Studies were excluded whether the inclusion criteria were not met.

### 2.2. Data Extraction and Quality Assessment

Data extraction forms were used to collect data from eligible studies. The following details were collected from each study: study characteristics (publication date, study location, study type, study sample, sampling method), population (patient characteristics, disease status, treatment status, psoriatic area severity index (PASI) score, details of comparison group), and outcome measures (serum IL-33, blood collection method used, methods used for assessing serum IL-33 levels). The quality of the included studies was assessed using NIH Quality Assessment Tool [21].

### 2.3. Statistical Analyses

The data compiled from the included studies were collated using three methods. An initial narrative analysis was conducted to describe the included studies. Vote counting was performed to explore the relationship between serum IL-33 levels among psoriatic and healthy participants and to explore the relation between IL-33 levels and PASI score/disease severity. The statistically significant threshold for vote counting was identified as $p < 0.05$. For serum IL-33 levels, the data collected were pooled to calculate summary estimates. We calculated the standardized mean difference and reported the pooled results with 95% CI. Heterogeneity was calculated using $I^2$ statistics and $I^2$ statistics of >70% were taken as high heterogeneity. The fixed-effects model was considered in case of no heterogeneity and random effects were considered in case of high heterogeneity ($I^2 > 70\%$). Publication bias was also explored using funnel plots. Meta-analysis was conducted using comprehensive meta-analysis (CMA) software [22]. A sensitivity analysis was performed by excluding the studies with a high risk of bias.

## 3. Results

### 3.1. Study Characteristics and Search Results

The search generated 830 potential records (Figure 1) and after removing duplicates, 814 records were screened for titles and abstracts. Only seven eligible studies were identified for full-text eligibility assessment. Among these, we only included six studies in the meta-analysis, as one of the studies was a protocol study and hence was excluded.

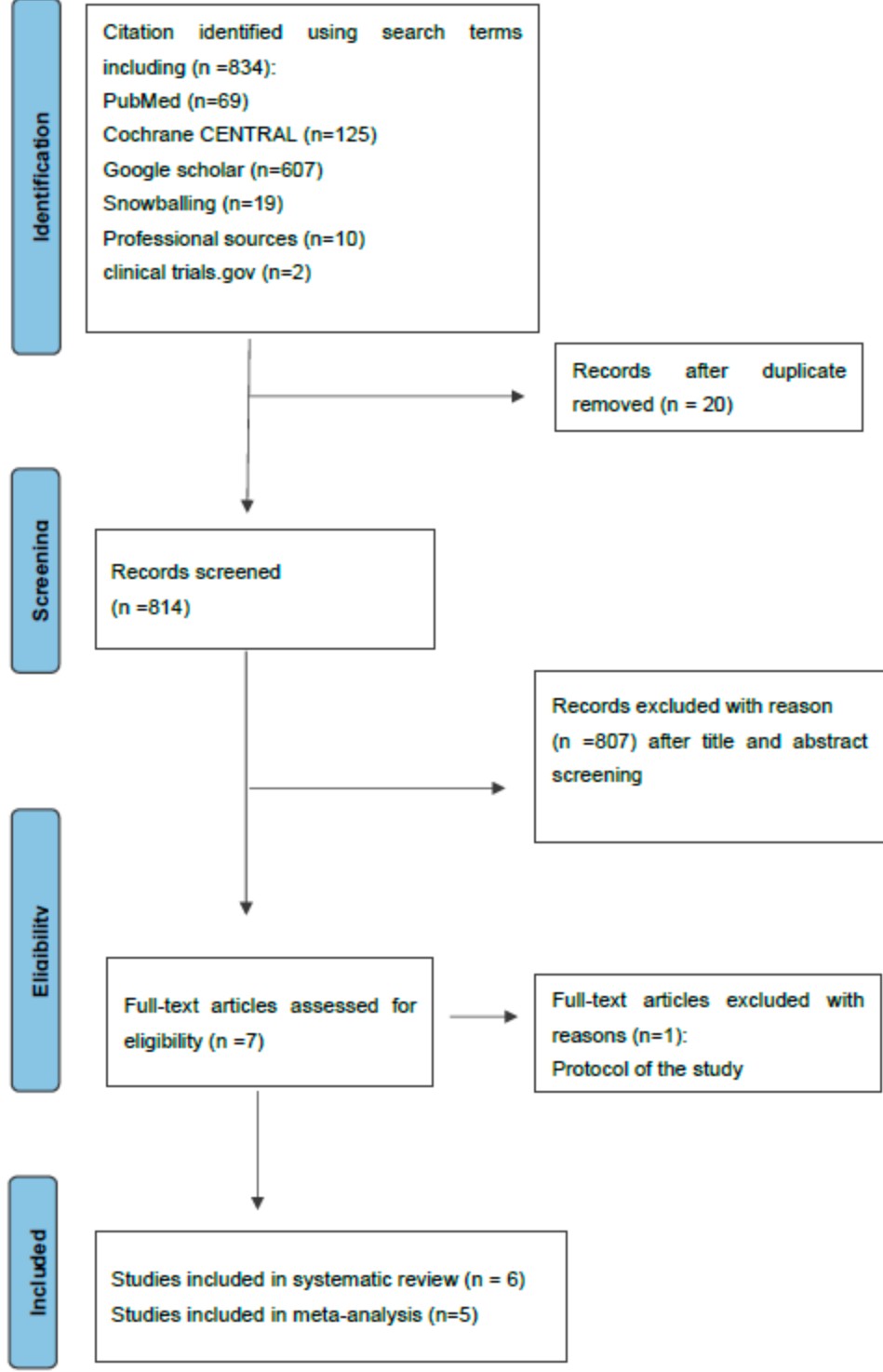

**Figure 1.** Flow chart of identification of studies included.

All included studies were observational cohort studies comparing two groups (Psoriatic Group and Control Group) conducted across different countries: China [11,15], Czech Republic [23], Iran [9], Japan [7] and Jordan [8]. Characteristics of the Psoriatic Group: All participants included in the psoriatic groups had a known diagnosis of psoriasis. The participant age profile included both the children and adults. The age range of the participants was reported as between 15–80 years (Median: 43 years) in one study [22] and between 32–63 years in another study (Median: 53 years) [11]. While the mean age reported across the study varied from 35 years (5–71 years) [8,9] to 46.6 years (27–61 years) [7]. Regarding treatment status, some studies excluded patients who have received treatment weeks before the study was conducted [15,23].

Whereas one study included patients receiving topical steroids and/or topical vitamin D3 but excluded patients with any systemic therapy [7], the remaining three studies [8,9,11] did not report the treatment status of the participants at the time of enrollment. PASI score among the psoriatic group was measured in five studies [7–9,11,23]. However, only three studies reported it [9,11,23]. Characteristics of the Control Group: Healthy individuals without any familial history of psoriasis were considered as controls in two studies [9,15]. Healthy blood donors were considered as the control in one study [23] while the health status of controls was not reported in the remaining studies [7,8,11]. Three studies [7,9,23] reported the age of the participants in the control group where the median age was reported to be 46 years (range: 21–65 years) in one study [23] and the mean age of 39 years (range: 33–45 years) [7] and 30.17 [9] years was reported in two studies. Of the included studies, one study [8] matched the psoriatic and control group by age and gender, one study was matched by age [9] while the remaining studies [7,11,15,23] reported no significant difference between the two groups at baseline. In terms of serum IL-33 levels, one study [23] reported the median serum level of IL-33, while three studies [7–9] reported the mean serum IL-33 level. Two studies [11,15] graphically represented the distribution of IL-33 levels using Whisker and Plot diagrams, and thus the mean values were visually approximated from the graphs. Detailed characteristics of included studies are shown in Table 2.

**Table 2.** Characteristics of the included studies [7–9,11,15,23].

| Study Name, Year of Publication and Location | Type of Study | Psoriatic Group | | | Control Group | | Outcome Measure |
| --- | --- | --- | --- | --- | --- | --- | --- |
| | | IL-33 Level (pg/mL) | PASI$_x$ SCORE | Sample Size (*n*) | IL-33 Level (pg/mL) | Sample Size (*n*) | Methods Used for Measuring Serum IL-33 |
| Borsky 2020 [23] *Czech Republic* | Observational cohort study | 4.890 (median) IQR * (2.94–7.96) | Median 17.4 | Total: 63 Female: 47.62% Male: 52.38% | 3.11 (median) IQR * (2.16–5.20) | Total: 95 Female: 47.4% Male: 52.6% | ELISA + |
| Chen 2020 [15] *China* | Observational cohort study | 0.35 (mean) | Not reported | 25 | 0.15 (mean) | 15 | ELISA |
| Mitsui 2014 [7] *Japan* | Observational cohort study | Mean 586 pg/mL (234–3900) | Measured but not reported | Total: 15 Female: 6.7% Male: 93.3% | Mean 87.7 pg/mL (60–197) | Total: 17 Female: 41% Male: 59% | ELISA |
| Bodoor 2020 [8] *Jordan* | Observational cohort study | Mean 29.55 pg/mL | Measured but not reported | 59 | Mean 44.88 pg/mL | 49 | ELISA |
| Sehat 2018 [9] *Iran* | Observational cohort study | Mean 19.21 ± 9.43 pg/mL | Mild (<11): 38 (80.9%) Moderate (11–19): 4 (8.5%) Severe (>19): 5 (10.6%) | 47 | Mean 19.30 ± 6.58 | 47 | ELISA |
| Li 2017 [11] *China* | Observational cohort study | 95 (mean) | Median: 3 Range (0.3–7.2) | Total: 20 Female:30% Male: 70% | 0 (mean) | Total: 20 Female: 35% Male: 65% | Not reported |

* Inter Quartile Range; $_x$ Psoriasis Area Severity Index; + Enzyme-Linked Immunosorbent Assay.

### 3.2. Quality of the Individual Studies

Since all the included studies were observational, the quality of the individual studies was assessed using the NIH quality assessment tool for observational cohort studies, which consisted of 14 questions/criteria pertaining to the study's internal validity [21]. The quality of the study was determined based on the number of questions that were answered 'yes' (75%: the high-quality or low risk of bias; >50–70%: the moderate quality or moderate risk of bias; and <50%: the low-quality or high risk of bias). The overall quality of the three included studies were considered to be moderate [7–9] and the rest of the studies were of low quality [11,15,23]. Please refer to the Appendix A.

### 3.3. Serum IL- 33 and Relation with Disease Severity or PASI Score

Four studies [7,9,11,23] reported the relationship between IL-33 levels and disease severity depicted by the PASI score. Among these, only one study [9] reported that there was a significant association between IL-33 serum levels and disease severity, where the $p < 0.001$.

### 3.4. Quantitative Data Synthesis: Serum IL-33 and Psoriasis

We included five studies in the meta-analysis, as one study [8] reported the median serum IL-33 and hence was not included. The mean IL-33 levels obtained were compiled and the pooled effect is depicted in Figure 2. Substantial heterogeneity in findings was found, as indicated by $I^2 = 86.347\%$ and $p < 0.000$, hence we used the random-effects model. The pooled standard difference in the mean was 0.340 (95% CI −0.308 to 0.988) which suggested that no statistically significant difference in the serum IL-33 levels between the psoriatic patients and healthy controls.

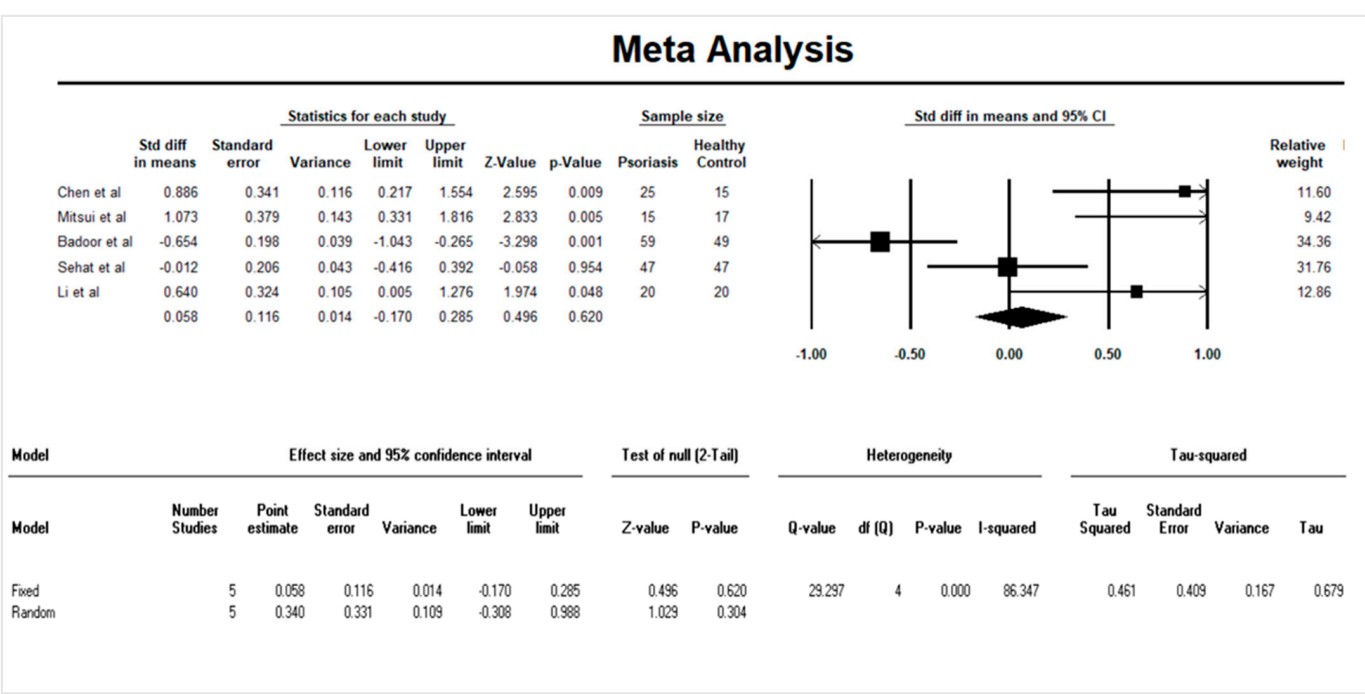

**Figure 2.** Forest plot of serum IL-33 and Psoriasis [7–9,11,15].

### 3.5. Sensitivity Analysis

A sensitivity analysis was performed by excluding the studies with a high risk of bias [11,15,23]. No statistically significant difference in the serum IL-33 (random effect model: standard mean difference 0.078 and 95% confidence interval (CI) −0.743 to 0.900) levels between psoriatic patients and healthy controls were observed when studies with a moderate risk of bias were only included. Please see Appendix A for the figure.

### 3.6. Publication Bias

To evaluate the potential for publication bias, a funnel plot was created to depict the standard error of the mean difference. The funnel plot did not resemble a funnel shape and the majority of the studies were concentrated at the bottom, on one side of the mean, indicating the presence of bias. Please check Appendix A for the figure.

## 4. Discussion

This meta-analysis investigated the correlation between serum IL-33 and psoriasis across five studies. The findings suggested no statistically significant difference in serum IL-33 between the psoriatic patients and healthy controls which included studies showing varied results. The range of IL-33 serum levels in both psoriatic patients and controls within the studies was inconsistent. Some of the included studies reported higher concentrations of the cytokine in the serum [7] and some [9] reported equal concentrations among psoriatic groups in comparison to healthy controls. This could be attributed to the ELISA kit used for measuring the serum IL-33 level or the severity of the disease at the time when the blood sample was collected. The baseline serum concentration of IL-33 among the various populations has not been yet studied, which could also be the reason behind the inconsistency. In addition, various systemic treatment modalities have an effect on IL-33 levels. Treatment with TNF-alpha1 such as adalimumam, etanercept and infliximab reduced Th17 responses, which may result in reduced IL-33 levels [24]. This is further supported by three studies that reported reduced IL-33 levels following TNF-alpha1 therapy [7,24–26]. A similar effect was visible among patients that received methotrexate therapy, which indirectly suppresses IL-17, TNF-alpha and INF-gamma [16]. Most studies included in this systematic review excluded participants from systematic therapy within two weeks following blood collection day. However, these treatments may have a long-term effect and could have resulted in the inconsistency in the serum IL-33 levels.

Contrary to the findings from this review, a previous literature review [19] which explored the possible roles of IL-33 in the pathogenesis of psoriasis concluded that among psoriatic patients, lesion IL-33 levels were consistently increased in all included studies in comparison to healthy controls. This analysis also studied the association between IL-33 levels and PASI scores as a representation of disease severity. Only one included study found the relationship to be statistically significant [9].

We considered some limitations during the systematic study. First, the number of participants considered in both the psoriatic group and control group across the studies was relatively small. Second, substantial heterogeneity was noted across the studies, which may have been due to the age of the participants, the effect of prior systemic therapy, ELISA testing kits, and blood collection and storage methods used. For some studies included in the review, serum IL-33 levels were represented graphically and not reported in the text. The data was extracted and approximated from the graphs, which may have influenced the estimate obtained. Further, studies included in the analysis were of moderate quality, which might have affected the results of this analysis. Furthermore, some previous studies have shown a relationship between IL-33 and psoriasis but most of them did not qualify according to our research. For example, we did not take into consideration (a) any animal studies, (b) lab-based studies, (c) any other markers mixed together, or (d) any other disease mixed together. These caveats in our study have led to a typical conclusion that we believe can open new discussions regarding the relationship between IL-33 and psoriasis.

To our understanding, there are not any other reviews about the meta-analysis of the data regarding serum IL-33 levels and psoriasis. Through meta-analysis, we combined the smaller studies, essentially making them into one big study, which helped us show an effect. Additionally, our meta-analysis of the data helped, increasing the accuracy of the results. In this, we attempted to eliminate the ambiguities that were persistent previously regarding serum IL-33 and psoriasis. Furthermore, we also conducted a sensitivity analysis to confirm the robustness of the results obtained.

## 5. Conclusions

This systematic review and meta-analysis concluded that there were no significant differences between serum IL-33 levels in the psoriatic population in comparison to healthy controls. Clinically, with the evidence that is currently available, it is difficult to suggest the use of serum IL-33 levels as a diagnostic marker. These findings add to the evidence; however, there is still a gap in research as there is a dearth of high-quality observational research with uniform research and investigational methodology that measure the serum IL-33 levels in a larger population. Some of the previous studies showed that IL-33 is increased in psoriasis. However, our results showed no significant increase in the IL-33 level. This might be because we did not include some of the parameters that other researchers used in their research. For example, we did not take into consideration (a) any animal studies, (b) lab-based studies, (c) any other markers mixed together, or (d) any other disease mixed together. In our study, vote counting was performed to explore the relation of serum IL-33 levels among psoriatic and healthy participants and to explore the relation of IL-33 levels and PASI score/disease severity. Statistically significant threshold for vote counting was identified as $p < 0.05$. The statistical analysis was carried out by using SPSS software. Heterogeneity among the observational studies was identified and a meta-analysis was undertaken using the effect model. $I^2$ statistics and overlap of the individual forest plots helped to determine the heterogeneity of the studies. $I^2$ statistics of >70% were taken as high heterogeneity. Publication bias among these studies was also explored using funnel plots. Meta-analysis was conducted using comprehensive meta-analysis (CMA) software.

**Author Contributions:** K.K.: Conceptualization, Methodology, Formal analysis, Investigation, Project administration, Writing—original draft; H.H.: Methodology, formal analysis; T.M.: Literature search, data extraction and management; S.K.: Literature search, data extraction and management; S.S.: Writing—review and editing; S.K.S.: Literature search, review, and editing; X.S.: Conceptualization, methodology, formal analysis, investigation, supervision; Z.T.: Conceptualization, methodology, formal analysis, investigation, supervision, project administration. All authors have read and agreed to the published version of the manuscript.

**Funding:** This work was funded by the Science and Technology Research Project of Henan Province grant no. 202102310184.

**Informed Consent Statement:** Informed consent was obtained from all subjects involved in the study.

**Acknowledgments:** We are grateful to the faculty of Master of clinical medicine (Dermatology), at Xinxiang Medical University for their support throughout the preparation of this review. We would also like to thank the professionals who supported the literature search process of this review.

**Conflicts of Interest:** The authors declare no conflict of interest.

## Appendix A

**Table A1.** Description of the quality assessment of the included studies.

| Study | Q1 | Q2 | Q3 | Q4 | Q5 | Q6 | Q7 | Q8 | Q9 | Q10 | Q11 | Q12 | Q13 | Q14 | Yes (*n*,%) | Quality |
|---|---|---|---|---|---|---|---|---|---|---|---|---|---|---|---|---|
| Borsky et al [23] | Yes | Yes | Not reported | Yes | Not reported | Yes | Yes | No | Yes | No | No | Not reported | Not reported | Not reported | 6.43% | Low |
| Chen et al [15] | Yes | No | Not reported | Yes | Not reported | Yes | Yes | No | Yes | No | Yes | Not reported | Not reported | Not reported | 6.43% | Low |
| Mitsui et al [7] | Yes | Yes | Not reported | Yes | Not reported | Yes | Yes | No | Yes | No | Yes | Not reported | Not reported | Not reported | 7.50% | Moderate |
| Badoor et al [8] | Yes | No | Not reported | Yes | Not reported | Yes | Yes | No | Yes | No | Yes | Not reported | Not reported | Yes | 7.50% | Moderate |
| Sehat et al [9] | Yes | No | Not reported | Yes | Not reported | Yes | Yes | Yes | Yes | No | Yes | Not reported | Not reported | Yes | 8.57% | Moderate |
| Li et al [11] | Yes | No | Not reported | Yes | Not reported | Yes | Yes | No | Yes | No | Yes | Not reported | Not reported | Not reported | 6.43% | Low |

Q1. Was the research question or objective in this paper clearly stated?
Q2. Was the study population clearly specified and defined?
Q3. Was the participation rate of eligible persons at least 50%? Were all the subjects selected or recruited from the same or similar populations (including the same time period)?
Q4. Were inclusion and exclusion criteria for being in the study prespecified and applied uniformly to all participants?
Q5. Was a sample size justification, power description, or variance and effect estimates provided?
Q6. For the analyses in this paper, were the exposure(s) of interest measured prior to the outcome(s) being measured?
Q7. Was the timeframe sufficient so that one could reasonably expect to see an association between exposure and outcome if it existed?
Q8. For exposures that can vary in amount or level, did the study examine different levels of the exposure as related to the outcome (e.g., categories of exposure, or exposure measured as continuous variable)?
Q9. Were the exposure measures (independent variables) clearly defined, valid, reliable, and implemented consistently across all study participants?
Q10. Was the exposure(s) assessed more than once over time?
Q11. Were the outcome measures (dependent variables) clearly defined, valid, reliable, and implemented consistently across all study participants?
Q12. Were the outcome assessors blinded to the exposure status of participants?
Q13. Was loss to follow-up after baseline 20% or less?
Q14. Were key potential confounding variables measured and adjusted statistically for their impact on the relationship between exposure(s) and outcome(s)?

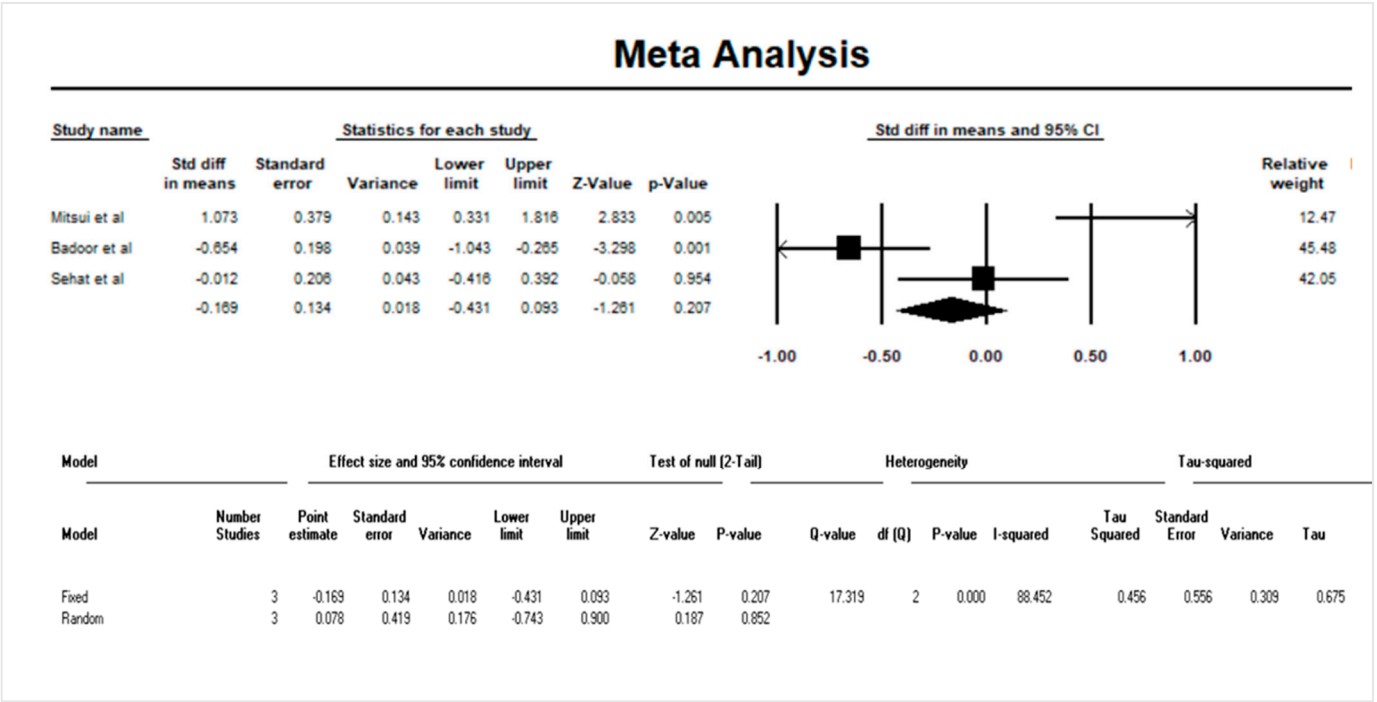

**Figure A1.** Sensitivity analysis [7–9].

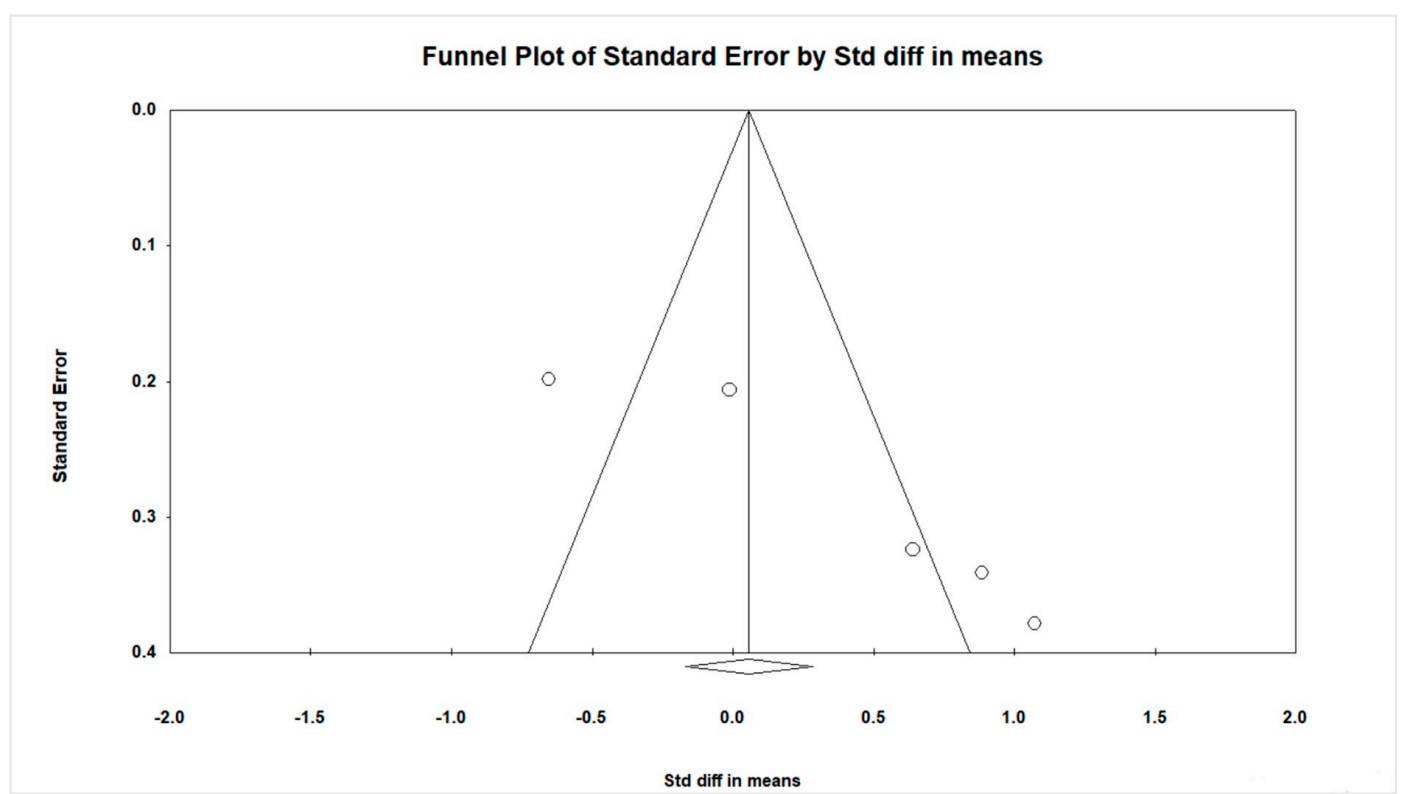

**Figure A2.** Funnel Plot.

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
