# Peer review of "The Correlation between Interleukin 33 and Psoriasis: A Systematic Review and Meta-Analysis"

_dermato, doi:10.3390/dermato3010002_

Round 1

Reviewer 1 Report

This paper is the first systematic review and meta-analysis of the correlation between psoriasis patients and serum IL-33 and is very informative.

I have no problem with the construction of the theory or the methods used, and I think the paper is well worth publishing, but I have two questions.

1) The article does not discuss the expression of IL-33 in diseases other than psoriasis. I believe that many readers lack basic knowledge about IL-33, so the author should insert basic knowledge such as IL-33 is involved in allergic diseases, rheumatism, IBD, SLE, Alzheimer's disease, etc. Since IL-33 is also detected in RA, IBD, etc., it would be easier to accept the possibility of its involvement in psoriasis.

2) This is more complicated and difficult to consider, but the background of the psoriasis patients cited (complications such as diabetes, habits: smoking, alcohol consumption, etc.) is not taken into account. Smoking, for example, increases IL-33 in airway cells and decreases IL-33 in serum Biochimica et biophysica acta 2014Sep01 Vol. 1842 issue(9).

Since psoriasis is a disease in which disease activity is influenced by lifestyle, it may be necessary to further take into account background factors such as smoking, obesity, and hyperlipidemia in the analysis. Since it is difficult to consider such factors at this stage, how about including them in the discussion as the next issue?

Author Response

Author Responses

Reviewer 1:

This paper is the first systematic review and meta-analysis of the correlation between psoriasis patients and serum IL-33 and is very informative. I have no problem with the construction of the theory or the methods used, and I think the paper is well worth publishing, but I have two questions.

Author response: We sincerely thank the reviewer. We are glad that the reviewer enjoyed reading the manuscript.

  • The article does not discuss the expression of IL-33 in diseases other than psoriasis. I believe that many readers lack basic knowledge about IL-33, so the author should insert basic knowledge such as IL-33 is involved in allergic diseases, rheumatism, IBD, SLE, Alzheimer's disease, etc. Since IL-33 is also detected in RA, IBD, etc., it would be easier to accept the possibility of its involvement in psoriasis.

Author response: Thank you for bringing this to our attention. We have added some information about IL-33 in the manuscript. It has been highlighted with yellow color.

  • This is more complicated and difficult to consider, but the background of the psoriasis patients cited (complications such as diabetes, habits: smoking, alcohol consumption, etc.) is not taken into account. Smoking, for example, increases IL-33 in airway cells and decreases IL-33 in serum Biochimica et biophysica acta 2014Sep01 Vol. 1842 issue(9).

Author response: Thank you for this constructive comment. We have modified our manuscript by adding the required citations in page 3 (line 90).

Since psoriasis is a disease in which disease activity is influenced by lifestyle, it may be necessary to further take into account background factors such as smoking, obesity, and hyperlipidemia in the analysis. Since it is difficult to consider such factors at this stage, how about including them in the discussion as the next issue?

Author response: We sincerely thank the reviewer for this insightful comment. We agree with the statement. However, the scope of this manuscript is to study the meta-analysis of the correlation between psoriasis patients and serum IL-33.

Reviewer 2 Report

The authors describe a meta-analysis of 5 clinical studies with the aim to determine whether the serum levels of IL-33 correlated with disease severity of plaque psoriasis. The authors find that though a number of the individual studies show significant differences of serum levels between plaque psoriasis patients and healthy controls, the pooled studies showed no significant difference in IL-33 serum levels.

I have a number of comments and questions below:

-The authors do not discuss the IL-33 serum levels and its relation to disease severity as outlined in their aim, rather discuss whether IL-33 is more significant in psoriasis patients than healthy controls.

-It is also not clear why other types of psoriasis (guttate, inverse psoriasis) were included in the database search. A difference in these clinical types of psoriaitc disease would surely confound any meaningful analysis further. Incidentally "psoriasis vulgaris" was not included in their search terms though plaque psoriasis was what in the end they had decided to study.

-It is unclear how meaningful the findings are. A number of the studies show significant differences in IL-33 serum levels and also that IL-33 decreased in response to anti-TNF treatment. 2 of the studies included in the review were studies that included patients who were being treated with anti-TNF blockers or other immunomodulators for their disease. In these 2 studies, patients were included if they hadn't treatment in 4 or 2 weeks before the respective studies. It is probable that the IL-33 serum levels would be lower in these included patients, such a short time after treatment.

-The inclusion criteria were not very stringent. There are many factors in the studies that would skew the data, for example the treatment that patients were receiving. Including children and adults in the same study might not be advisable. Children probably have an underlying genetic link to their disease and might have much more severe disease than adults.

minor comments:

- the abstract needs to be checked by a native English speaker though the rest of the article the English was better.

- there are some typos (e.g. line 70 "Koebner", line 64 "TNF-a")

- there are inconsistencies with text, eg. Th2 (subscript 2) on line 60, differs to the way Th17 is written. also IL33 line 158

- in many places in text there are 2 spaces between words, e.g. lines 59, 84, 91, 93, 193, 243, 260

- Table 1 should include the reference number, as it is a bit confusing trying to correlate the studies with the text. Indeed, the authors themselves claim on line 200 that reference 8 (Badoor et al) is left out of the meta-analysis, however, it is actually reference 22 that is excluded (Borsky et al).

Author Response

                               Author Response

Reviewer 2:

The authors describe a meta-analysis of 5 clinical studies with the aim to determine whether the serum levels of IL-33 correlated with disease severity of plaque psoriasis. The authors find that though a number of the individual studies show significant differences of serum levels between plaque psoriasis patients and healthy controls, the pooled studies showed no significant difference in IL-33 serum levels.

 Author response: We sincerely thank the reviewer for taking the time to provide insightful suggestions.

I have a number of comments and questions below:

-The authors do not discuss the IL-33 serum levels and its relation to disease severity as outlined in their aim, rather discuss whether IL-33 is more significant in psoriasis patients than healthy controls.

Author response: Thank you for the comment. We had only limited our study to the IL-33 level with psoriasis. In this regard, the extraction forms were used to collect data from the eligible studies. The following details were collected from each study: study characteristics (like publication date, study location, study type, study sample, and sampling method), population (like patient characteristics, disease status, treatment status, psoriatic area severity index (PASI) score, details of the comparison group), and outcome measures (like, serum IL-33, blood collection method used, methods used for assessing serum IL-33 levels).

-It is also not clear why other types of psoriasis (guttate, inverse psoriasis) were included in the database search. A difference in these clinical types of psoriaitc disease would surely confound any meaningful analysis further. Incidentally "psoriasis vulgaris" was not included in their search terms though plaque psoriasis was what in the end they had decided to study.

Author response: We had conducted the search until 30th September 2021, using the search terms: (interleukin 33 OR IL-33 OR IL33) and (psoriasis or palmo-plantaris pustulosis or pustular psoriasis of palms and soles or pustulosis palmaris et plantaris or pustulosis of palms and soles or plaque psoriasis or guttate psoriasis or inverse psoriasis or pustular psoriasis or erythrodermic psoriasis). We limited the search to only human study.  We also searched the reference list of published systematic reviews (snowballing) and contacted professionals in the relevant field. The inclusion criteria were: (a) Population: adults (aged ≥18 years) and children (<18 years); (b) Disease: Psoriasis; (c) Comparator/Control: healthy adults and children; (d) Outcome measure: serum IL-33. Studies were excluded if the inclusion criteria were not met.

-It is unclear how meaningful the findings are. A number of the studies show significant differences in IL-33 serum levels and also that IL-33 decreased in response to anti-TNF treatment. 2 of the studies included in the review were studies that included patients who were being treated with anti-TNF blockers or other immunomodulators for their disease. In these 2 studies, patients were included if they hadn't treatment in 4 or 2 weeks before the respective studies. It is probable that the IL-33 serum levels would be lower in these included patients, such a short time after treatment.

Author response: Some of the included studies reported higher concentrations of the cytokine in serum and some reported equal concentrations among psoriatic groups in comparison to healthy controls. This could be attributed to the ELISA kit used for measuring the serum IL-33 level or the severity of the disease at the time when the blood sample was collected. The baseline serum concentration of IL-33 among the various population is not yet studied which could also be the reason behind the inconsistency.

Treatment with anti-TNF-alpha1: such as adalimumab, etanercept, and infliximab reduced Th17 responses which may result in reduced IL-33 levels. This is further supported by three studies that reported reduced IL-33 levels following TNF-alpha1 therapy. A similar effect was visible among patients that received methotrexate therapy which indirectly suppresses IL-17, TNF-alpha and INF-gamma. Most studies included in this systematic review excluded participants from systematic therapy within two weeks following blood collection day. However, these treatments may have a long-term effect and could have resulted in the inconsistency in serum IL-33 levels.

-The inclusion criteria were not very stringent. There are many factors in the studies that would skew the data, for example the treatment that patients were receiving. Including children and adults in the same study might not be advisable. Children probably have an underlying genetic link to their disease and might have much more severe disease than adults.

Author response: The main aim of meta-analysis and the meaningfulness of the study was to know the correlation between IL-33 and psoriasis from the published journals till 2021. Our study showed that there are no significant changes between healthy control and psoriatic patient. Some publication bias was also found. We just did the meta-analysis that our authors found and got the result. We assumed this type of result and conclusion, as a very low quantity of research articles, was published about IL-33 and psoriasis.

minor comments:

- the abstract needs to be checked by a native English speaker though the rest of the article the English was better.

Author response: Thank you. We asked native English speaker to go through the whole manuscript. We have changed as required as highlighted.

- there are some typos (e.g. line 70 "Koebner", line 64 "TNF-a")

Author response: Thank you. We have modified them.

- there are inconsistencies with text, eg. Th2 (subscript 2) on line 60, differs to the way Th17 is written. also IL33 line 158

Author response: Thank you. We have removed the inconsistencies.

- in many places in text there are 2 spaces between words, e.g. lines 59, 84, 91, 93, 193, 243, 260

Author response: Thank you. We have made uniformity in the line spaces.

- Table 1 should include the reference number, as it is a bit confusing trying to correlate the studies with the text. Indeed, the authors themselves claim on line 200 that reference 8 (Badoor et al) is left out of the meta-analysis, however, it is actually reference 22 that is excluded (Borsky et al).

Author response: Thank you for pointing out this error. We have modified the manuscript following the reviewer’s suggestion.

Round 2

Reviewer 1 Report

I believe this paper has been appropriately revised and is worthy of publication.

Author Response

Reviewer 1:

I believe this paper has been appropriately revised and is worthy of publication.

Author response: We sincerely thank the reviewer for the positive response.

Reviewer 2 Report

The authors haven't really addressed my concerns or even discussed them in the manuscript discussion. 

I haven't been convinced that this is a significant finding for the scientific community and that IL-33 doesn't play a role in psoriasis. Contrasting literature would suggest otherwise and a meta-analysis on 5 very different clinical studies might be leading to an incorrect conclusion.

Additionally, the language should have been improved but I find still many typos. I am disappointed that the authors didn't take the reviewers' in put as an opportunity to improve the manuscript.

Author Response

We sincerely thank the reviewer for the positive comments. Due to the limited study of cytokines   IL-33 and its relationship with psoriasis, we got very little data. So, such a result was established. As the reviewer said, before doing research, we all had the same mindset that IL-33 is increased in psoriasis. However, the result showed no significant increase in IL- 33 level. Here are some reasons behind this:

There are many articles published showing the relationship between IL-33 and psoriasis but most of them didn't qualify according to our research. For example, we didn’t take into consideration of a) any animal studies, b) lab-based studies, c) any other markers mixed together, or d) any other disease mixed together.

We took a significantly longer time to address the concern brought by the reviewer. We have included the tables below that illustrates the criteria that we have considered in our study. Furthermore, we have shown what was included and what was excluded in the study. In our study:

1) The data compiled from the included studies were assimilated using three methods.

2) An initial narrative analysis was planned to describe the included studies.

3) Vote counting was performed to explore the relation of serum IL-33 levels among psoriatic and healthy participants and to explore the relation between IL-33 levels and PASI score/ disease severity.

4) Statistically significant threshold for vote counting was identified as p < 0.05. 

5) Statistical analysis was carried out by using SPSS software.

6) Heterogeneity among the observational studies was identified and a meta-analysis was undertaken using the effect model.

7) I2 statistics and overlap of the individual forest plots helped to determine the heterogeneity of the studies.

8) I2 statistics of  >70% were taken as high heterogeneity. Publication bias among these studies was also explored using funnel plots. Meta-analysis was conducted using comprehensive meta-analysis (CMA) software.

  • All included studies were observational cohort studies comparing two groups.
  • Three studies were published in the year 2020 and one each in 2014, 2018, and 2017.
  • Two studies were conducted in China (Chen et al, April 2020 and Li et al, May 2017) and one each in the Czech Republic, Iran, Japan, and Jordan.

We have attached a separate file to address the reviewer's comments. 

Round 3

Reviewer 2 Report

The authors have answered my questions. However, I do still feel that a meta-analysis only performed on 5 articles (from 800+ articles) can in itself be a misrepresentation though I do agree with the authors criteria.

I would like, however, that the authors discuss the caveats of their work in the discussion or conclusion,  i.e. including both adult and children patient studies, only 5 manuscripts tested, etc.

Author Response

Reviewer 2: The authors have answered my questions. However, I do still feel that a meta-analysis only performed on 5 articles (from 800+ articles) can in itself be a misrepresentation though I do agree with the authors criteria.

I would like, however, that the authors discuss the caveats of their work in the discussion or conclusion,  i.e. including both adult and children patient studies, only 5 manuscripts tested, etc.

Author Response: We sincerely thank the reviewer for their insightful comments. The comments have really helped in increasing the significance of our manuscript. By following the reviewer’s suggestion, we have added the caveats of our work in page 12, line 265-278.
